

# Generative named entity recognition framework for Chinese legal domain

Xingliang Mao[1], Jie Jiang[2], Yongzhe Zeng[3], Yinan Peng[3], Shichao Zhang[3] and Fangfang Li[3]

[1] School of Digital Media Engineering and Humanities, Hunan University of Technology and Business, Changsha, Hunan, China
[2] College of Systems Engineering, National University of Defense Technology, Changsha, Hunan, China
[3] School of Computer Science and Engineering, Central South University, Changsha, Hunan, China

## ABSTRACT

Named entity recognition (NER) is a crucial task in natural language processing, particularly challenging in the legal domain due to the intricate and lengthy nature of legal entities. Existing methods often struggle with accurately identifying entity boundaries and types in legal texts. To address these challenges, we propose a novel sequence-to-sequence framework designed specifically for the legal domain. This framework features an entity-type-aware module that leverages contrastive learning to enhance the prediction of entity types. Additionally, we incorporate a decoder with a copy mechanism that accurately identifies complex legal entities without the need for explicit tagging schemas. Our extensive experiments on two legal datasets show that our framework significantly outperforms state-of-the-art methods, achieving notable improvements in precision, recall, and F1 score. This demonstrates the effectiveness of our approach in improving entity recognition in legal texts, offering a promising direction for future research in legal NER.

## INTRODUCTION

With the advancement of judicial intelligence, the application of artificial intelligence in legal contexts has gained significant attention. Named entity recognition (NER) is a critical task in natural language processing that identifies specific spans of text in legal documents according to predefined types of entities (*Tjong Kim Sang & De Meulder, 2003*). The demand for NER in legal texts is paramount, enabling the precise extraction of crucial details such as victims, defendants, and crime timings. This capability is vital not only for streamlining legal documentation processes but also for supporting intricate legal research and case analysis. Therefore, investigating the implementation of NER techniques within the legal domain is of substantial importance.

Various deep learning based methods have been proposed to handle the NER tasks, such as sequence tagging methods (*Panchendrarajan & Amaresan, 2018*; *Ju, Miwa & Ananiadou, 2018*; *Zhao et al., 2019a*; *Luo, Xiao & Zhao, 2020*), span-based methods (*Tan et al., 2020*; *Fu, Huang & Liu, 2021*; *Zaratiana et al., 2022*; *Liu, Fan & Liu, 2022*), hypergraph-based methods (*Lu & Roth, 2015*; *Wang & Lu, 2018*; *Katiyar & Cardie, 2018*),

Corresponding author
Fangfang Li, lifangfang@csu.edu.cn

*etc*. Although these existing methods have dramatically advanced the NER tasks in general domain, the following issues still exist in the NER tasks in the legal domain. Firstly, it is challenging to accurately assign entity types to entity spans in the legal domain because of the diverse and professional nature of entities. As shown in Fig. 1, S1 shows an example of the diversity and professionalism of entity types in the legal texts. In the S1, the entities related to "Money" in the text are categorized into more specific types in the NER task in the legal domain: "1,500 yuan" is the entity belonging to "Stolen Currency", "5,000 yuan" is the entity belonging to "Value of stolen item", and "2,000 yuan" is the entity belonging to "Profit", according to their judicial properties in the case. However, most existing methods have ignored learning the semantic information of entity types. Secondly, legal texts contain intricate and lengthy entities that require precise identification to maintain the integrity of the legal system. Consequently, predicting entity boundaries in NER leads to more errors. As shown in Fig. 1, S2 shows an example of intricate and lengthy entity in the legal texts. In S2, the description of the "location" entity is more detailed than that found in general domain text, making it challenging to recognize the entity's boundary accurately.

Accordingly, we propose a simple sequence-to-sequence framework to tackle the NER tasks in the legal domain. In our method, the sequence-to-sequence pre-trained model BART (*Lewis et al., 2020*) is used as the base model and an entity-type-aware module is designed at the encoder side to optimize the encoding capability of the model's encoder through the label-attention mechanism and contrastive learning, which helps the model to learn the information about the entity types and the differences between different entity types, and reduces the model's likelihood of incorrect prediction of entity types so that the model can better recognize legal entities with diversity and professionalism similar to those in the S1. Meanwhile, we formulate the NER task as a sequence generation task and utilize the decoder with copy mechanism to generate the results of entity recognition directly, so as to accurately recognize the intricate and lengthy entities in the legal texts similar to those in the S2. Finally, the method jointly optimizes the loss of the contrastive learning task at the encoder side and the loss of the entity recognition results of generation task at the decoder side through multi-task learning to obtain a model suitable for the task of recognizing named entities in legal texts. The main contributions of the article can be summarized as follows:

- We formulate the NER task in the legal domain as a sequence generation task and then propose a simple sequence-to-sequence framework.
- To reduce the entity type prediction errors in the NER task in the legal domain, an entity-type-aware module is designed to assist the model in learning richer semantic information about entity types and the differences of each entity type.
- The extensive experiments on the CAIL2021 and Drug datasets show that our model outperforms a collection of state-of-the-art models in the NER task in the legal domain. A detailed ablation study is also conducted to validate the effectiveness of these modules.

S1：被告人王某盗走了现金1500元和一部价值5000元的手机，后将该手机以2000元出售。

The defendant Wang Mou stole 1,500 yuan in cash and an Iphone worth 5,000 yuan, and then sold the phone for 2,000 yuan.

S2：民警在停在江岸区铭心街隧道出口附近的一辆吉利汽车内抓获正在吸毒的王某。

The police arrested Wang Mou, who was taking drugs, in a Geely car parked near the tunnel exit of Mingxin street, Jiang'an district.

| ■ Stolen Currency | ■ Value of stolen item | ■ Profit | ■ Location |
|---|---|---|---|

**Figure 1** **Examples of entities in legal texts.** S1 shows an example of the diversity and professionalism of entity types in the legal texts, and S2 shows an example of intricate and lengthy entities in the legal text.

# RELATED WORK

Named entity recognition (NER) is a fundamental task in natural language processing. With the advancement of research in deep learning, various deep learning based methods have been proposed to tackle the NER task in general domain and legal domain.

## Sequence tagging method

Sequence tagging method is the classic method to tackle the NER task, which views the NER task as a token-level classification task, assigns a tag for each token from a pre-designed tagging scheme and utilizes conditional random field (CRF) or tag sequence generation methods to decode and obtain the NER results. *Xiaofeng, Wei & Aiping (2020)* present an incorporating token-level dictionary feature method, making it easier to quantify the effect of dictionary features and decoupling dictionary features from the external dictionary during the training stage. *Li et al. (2021)* combines the ordered neurons long short-term memory (LSTM) network (*Shen et al., 2018*) with the BERT model and CRF layer to enhance model's ability of mining the hierarchy information of text, which can improve the effect of model in NER task in the legal domain. *Shi et al. (2022)* combines the bidirectional LSTM (BiLSTM) network with RoBERTa model and CRF layer to tackle Chinese NER task in legal domain, which achieves significant effect on legal NER. *An et al. (2022)* have designed an improved character-level representation approach to enhance the specificity and diversity of feature representations, and propose a multi-head self-attention based BiLSTM-CRF (MUSA-BiLSTM-CRF) model to tackle the Chinese clinical NER task, which achieves the best performance on two CCKS challenge benchmark datasets. *Deng, Lin & Lu (2023)* propose a multi-feature fusion model for Chinese named entity recognition that enhances BERT by fully utilizing information across radicals, words, and lexical levels, thereby addressing the inefficiencies in Chinese NER concerning the use of radicals and words, and achieve better F1 values on the Weibo dataset and OntoNotes4.0 dataset. *Wang et al. (2024)* propose a lexicon-enhanced method for recognizing Chinese long named entities that integrates a SkipWord-Lattice module, which resolves issues with word overlap, redundancy, and confusion, with a Word-Aware Attention module that enhances the interaction between character tokens and word tokens, and achieve state-of-the-art (SOTA) performance on four Chinese NER datasets.

## Span-based method

Span-based method views the NER task as a span-level classification task, which aims to find all possible entity spans in the text and assign entity type for each span. Compared to sequence labeling methods, span-based methods are better able to extract nested entities. *Tan et al. (2020)* propose a boundary enhanced neural span classification model, which views the NER task as boundary detection task and span classification task, and achieves the SOTA performance in terms of F1 on the ACE2004, ACE2005, and GENIA datasets. *Yu, Bohnet & Poesio (2020)* utilize the bi-affine mechanism to calculate span classification scores in NER task, which can enhance the ability of model for locating entity spans, and achieve SOTA performance on eight corpora. *Huang et al. (2022)* design a span selection framework with generative adversarial training, which demonstrates the excellent performance on four nested NER datasets. *Zhang & Chen (2023)* propose a simple yet effective span-based model with knowledge distillation to preserve memories and multi-Label prediction to prevent conficts in continual learning for NER. *Zheng et al. (2024)* propose a lexicon-free Chinese NER framework called SENCR that incorporates a boundary detector for boundary supervision, a span-convolutional network for better span representation and classifcation and a novel counterfactual rethinking strategy in inference for debiased boundary detection, and prove the effectiveness of SENCR on four Chinese NER datasets.

Recently, some studies have designed new architectures or incorporated different paradigms to tackle the NER task. *Li et al. (2020)* formulate the NER task as a machine reading comprehension task, and extract possible entity spans by answering the queries given the contexts. *Yan et al. (2021)* model the NER task as a sequence generation task, and obtain the NER results in a generative way. *Li et al. (2022a)* propose a joint training multi-task model for multi-answer types of reading comprehension task by adopting a feature-based paragraph extraction mechanism to extract more relevant sentences in the passage. *Li et al. (2022b)* predict the relation between word and word in the sentence by modelling the NER task as word-word relation classification, and then extract entities by decoding based on the relation between word and word. *Lu et al. (2023)* recast the NER task to a seq2seq task where a prefix language modelling objective is introduced to reduce the gap between pre-training and fine-tuning. *Shen et al. (2023)* propose a generative approach for NER that converts the task into a boundary denoising diffusion process, which achieves comparable or better performance on six nested and flat NER datasets. *Zhang et al. (2023a)* propose a novel Decomposing Logits Distillation (DLD) method, which is model-agnostic and easy to implement and enhances the model's ability to retain old knowledge and mitigate catastrophic forgetting. *Zhang et al. (2023b)* propose a de-bias contrastive learning based approach for multimodal named entity recognition (MNER), which studies modality alignment enhanced by cross-modal contrastive learning. *Mo et al. (2024)* introduce MCL-NER, a multi-view contrastive learning framework for cross-lingual NER that encompasses semantic and token-to-token relation contrastive learning, and constructs code-switched data by randomly replacing some phrases with the target counterparts for the semantic contrastive learning of the source and corresponding

code-switched sentence, and performs better than baselines by a large margin on the XTREME-40 and CoNLL benchmarks.

While these methods have significantly advanced the NER task in both general and legal domains, there is a compelling need for a more comprehensive exploration of NER in the legal texts. Existing approaches often overlook the incorporation of semantic information related to entity types. Given the diversity and professionalism of entity types within legal texts, most current methods struggle to accurately assign entity types to entity spans in the legal domain. Additionally, legal texts frequently contain intricate and lengthy entities, posing challenges for both sequence tagging and span-based methods in entity extraction. To address these issues, we have devised a simple sequence-to-sequence framework aimed at enhancing NER performance in the legal domain.

# METHODS

To tackle the NER task in the legal domain, we propose a sequence to sequence framework, which consists of two main components: the encoder with entity-type-aware module and the decoder with copy mechanism. The overall structure of our framework is shown as Fig. 2.

## Problem formulation

In our method, the NER task is translated into a sequence to sequence task, which can be formulated as follows: given an sentence of $n$ tokens $X = \{x_1, x_2, \ldots, x_n\}$ and an entity type set $Y = \{y_1, y_2, \ldots, y_r\}$, where $r$ is the number of entity types. We view $X$ as input sequence, and the target sequence is to convert entity spans in the $X$ to a span like "[entity span | $y_i$]", where $y_i \in Y$ and the entity span belongs to $y_i$.

## Comprehensive discussion

Seq2seq models are inherently suited for tasks involving complex entity structures, such as those found in legal texts where entities like case references, legal citations, or multi-component names span several words or phrases. These models excel in capturing contextual dependencies due to their attention mechanisms, which significantly improves accuracy in identifying and classifying entities based on the surrounding text. Differentiating between various legal entity types also requires a deep understanding of the domain's semantics, which seq2seq models address by incorporating entity-type-aware modules that use contrastive learning. This enhances the model's ability to distinguish between different types of entities effectively. Moreover, seq2seq models address the challenge of predicting entity boundaries by framing entity recognition as a generation task, allowing for more flexible and accurate extraction of nested or overlapping entities. They can also integrate legal-specific knowledge by pre-training on large corpora of legal documents or incorporating legal dictionaries and definitions into the training process. Finally, seq2seq models are adaptable and scalable; they can be fine-tuned with examples from specific areas of law or jurisdictions to handle variability across legal documents and scale well with the addition of more training data.
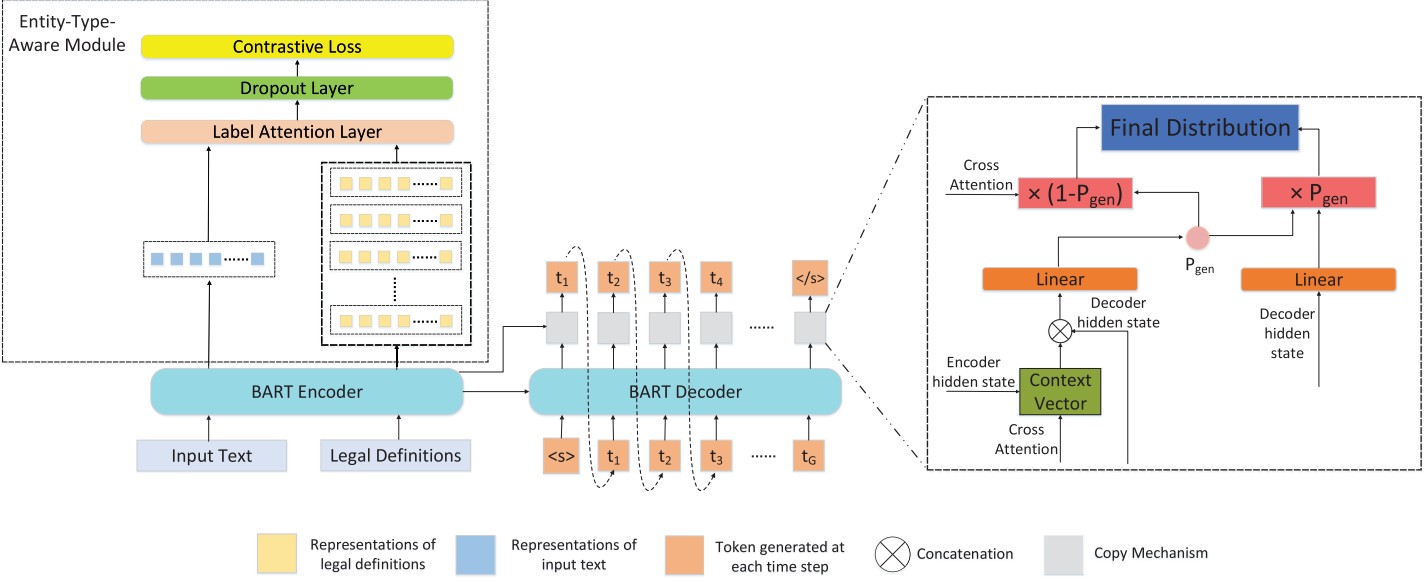

**Figure 2 The overall structure of our framework.** In the framework, the encoder is utilized to encode the input text and legal definitions, and then their representations are fed into entity-type-aware module, which can optimize the encoding capability of the encoder. Meanwhile, the output of the encoder is provided to the decoder for generating the entity recognition results. In the training phase, the contrastive learning task and entity recognition result generation task are jointly optimized. <s> and </s> are the predefined start-of-sentence and end-of-sentence tokens in BART.

## Encoder optimized by semantic information of entity types

In our method, the sequence to sequence pre-trained model BART is used as our base model. The BART encoder is responsible for encoding the text and captures the features of the text and providing the decoder with the semantic information to generate the entity recognition results. However, due to the diversity and professionalism of entity types in the legal text, it is hard for the model to assign each entity span with the entity type accurately. Therefore, an entity-type-awareness module is designed on the encoder to assist the model in learning the semantic information of entity types and the difference between each entity type, which can optimize the encoding capability of encoder and reduce the entity type prediction errors.

### Legal definitions construction

Inspired by *Li et al. (2020)*, to assist model in learning semantic information of each entity type, the legal definitions for each entity type are collected as prior information of each entity type, which are the comprehensive and professional definition of each entity type in the legislation. Examples of legal definitions are shown in Table 1.

To provide the prior information of entity types to improve the effect of our framework, we collect legal definitions[1] for each entity type, which are the comprehensive and professional definition of each entity type in the legislation. These legal definitions for each entity type on the CAIL2021 and Drug datasets are shown in Tables 1 and 2, respectively.

[1] We collect these legal definitions from Chinese Laws and Regulations Database.

**Table 1 Consolidated legal definitions for entity types in CAIL2021.**

| Entity type | Legal definition |
| --- | --- |
| Victim | The victim refers to a person whose legitimate rights and interests are directly infringed upon by an unlawful act. |
| Defendant | A defendant is a person accused of a crime in a criminal case and investigated for criminal responsibility. |
| Stolen items | Stolen items refer to items stolen by the defendant and belonging to the victim. |
| Organization | Organization refers to the specialized legal organs and organizations established in accordance with the law to formulate, implement and maintain the legal system. |
| Value of item | Value of item refers to the actual value of the item stolen by the defendant. |
| Crime time | Crime time refers to the time when the defendant committed the crime. |
| Crime location | Crime location refers to the place where the defendant committed the crime. |
| Stolen currency | Stolen currency refers to the tangible currency stolen by the defendant, including RMB, US dollar and other currencies. |
| Crime tool | Crime tool refers to all items used by the defendant to commit crimes, including various weapons or tools. |
| Profit | Profit refers to the benefit defendant obtained from the implementation of illegal and criminal activities. |

**Table 2 The legal definitions of each entity type in Drug.**

| Entity type | Legal definition |
| --- | --- |
| Defendant | The defendant refers to the person involved in a drug-related crime. |
| Drug | Drug refers to narcotic drugs and psychotropic substances that can make people dependent; such as ice, methamphetamine, *etc.* |
| Crime time | Crime time refers to the time when drug-related personnel take drugs or buy and sell drugs. |
| Crime location | Crime location refers to the place where the drug-related person committed the crime and the place where he was arrested. |
| Drug weight | Drug weight refers to the weight of the drugs involved in the case. |

### Pre-process

Similar to the text processing techniques that have been applied to the rumor detection task (*Li et al., 2023*), we preprocess the data for legal texts. First, special symbols in the data such as tabs, line breaks, *etc.*, that may affect the semantics of the text are removed. Second, due to the limited maximum length of text that can be processed by the BART model, we truncate the legal texts according to the structure and paragraphs of the case, and truncate a case into multiple texts of smaller lengths based on retaining the contextual semantic information, which facilitates the pre-trained language model to process the textual data in the subsequent process.

### Entity-type-aware module

An entity-type-aware (ETA) module is designed on the encoder to minimize the occurrence of entity type prediction errors. In our method, each token in the sentence has to be computed with the attention of each token of the legal definition, and different types of legal definitions will have different impacts on the semantics of the original sentence during the computation process, and there are differences in the vector information obtained in the end. Therefore, the ETA module aims to help the model learn semantic information about entity types and differences between different entity types by

contrastive learning. As a result, the encoder's encoding capability is optimized, and the performance of the model is enhanced.

Concretely, in the entity-type-aware module, given an input text $X = \{x_1, x_2, ..., x_n\}$ and a legal definition set $L = \{l_1, l_2, ..., l_r\}$, where $l_i(1 \leq i \leq r)$ is the legal definition of $i$-th entity type, we feed the input text and all legal definitions into the encoder and obtain their representations respectively:

$$h^{enc} = Encoder(X), \tag{1}$$

$$h_i^l = Encoder(l_i), \tag{2}$$

where $h_i^l$ is the embedding of legal definition of $i$-th entity type and $h^{enc}$ is the embedding of input text $X$. The embedding of each legal definition $h_j^l(1 \leq j \leq r)$ and input text are fed into a label attention layer, and the output of label attention layer is averaged to obtain the representation incorporating entity type semantic information $H_j$:

$$Attention(Q, K, V) = Softmax\left(\frac{QK^T}{\sqrt{d_k}}\right)V, \tag{3}$$

$$H_j = Mean(Attention(h^{enc}, h_j^l, h^{enc})), \tag{4}$$

Then a representation set $H = \{H_1, H_2, ..., H_r\}$ can be obtained. For $H_i \in H$ $(1 \leq i \leq r)$, after feeding $H_i$ into a dropout layer $\widehat{H}_i$, $H_i$ and the representation obtained are considered as a positive sample pair, and $H_i$ and $H_j(1 \leq j \leq r, j \neq i)$ are considered as negative sample pairs:

$$\widehat{H}_i = Dropout(H_i), \tag{5}$$

After constructing the positive and negative sample pairs, contrastive learning is utilized to minimize the distance between embeddings of positive pair and maximize the distance between embeddings of negative pairs, which can optimize the encoding capability of encoder, assist the model in learning the semantic information of entity types and the difference between each entity type and reduce the possibility of entity type prediction error. The InfoNCE loss (*Oord, Li & Vinyals, 2018*) is used as the loss of contrastive learning task, which is calculated as follows:

$$L_{con} = -\sum_{i=1}^{r} log \frac{e^{sim(H_i,\widehat{H}_i)/\tau}}{\sum_{j=1}^{r} f(j) \cdot e^{sim(H_i,H_j)/\tau}}, \tag{6}$$

$$f(j) = \begin{cases} 0 & j = i \\ 1 & j \neq i \end{cases}. \tag{7}$$

where $sim(\cdot)$ is the cosine similarity and $\tau$ is the temperature coefficient.

## Decoder with copy mechanism

To solve the problem of intricate and lengthy entities in the legal texts and avoid the problem of wrong generation, inspired by *See, Liu & Manning (2017)*, the decoder is

utilized to directly generate the entity recognition results, and the copy mechanism is utilized to restrict the process of decoding to ensure that the entity recognition results generated are consistent with corresponding parts of origin texts.

Formally, given an input text $X = \{x_1, x_2, ..., x_n\}$, the input text is fed into the encoder to obtain the embedding of the text, and the encoder's last hidden state is fed into the decoder for the entity recognition results generation. At the $c$-th step of the decoder, the encoder's last hidden state $h^{enc}$ and the previous output tokens $t_{1:c-1}$ are fed into the decoder, and the last hidden state of the decoder is calculated as follows:

$$h^{enc} = Encoder(X), \tag{8}$$

$$h^{dec}_c = Decoder(h^{enc}, t_{1:c-1}), \tag{9}$$

During the training and inference phase, the copy mechanism is utilized to restrict the process of decoding. The attention distribution of token $t_c$ and each token in the input sentence $X$ are first calculated:

$$g^c_i = v^T(W_h h^{enc}_i + W_s h^{dec}_c + b), \tag{10}$$

$$a^c = softmax(g^c), \tag{11}$$

where $v$, $W_h$, $W_s$, $b$ are the learnable parameters. Then the context vector $h^*_c$ is calculated as follows:

$$h^*_c = \sum_{i=1}^{n} a^c_i h^{enc}_i, \tag{12}$$

where $h^{enc}_i$ is the encoder hidden state of the token $x_i$, and $a^c_i$ is the attention score of token $t_c$ and token $x_i$. The vocabulary distribution $P^c_{vocab}$ is calculated as follow:

$$P^c = Softmax(Linear(h^{dec}_c)), \tag{13}$$

$$P^c_{gen} = Softmax(Linear([h^*_c; h^{dec}_c])), \tag{14}$$

$$P^c_{vocab} = p^c_{gen}P^c + (1 - p_{gen})\sum_{i=1}^{n} a^c_i, \tag{15}$$

where $P^c_{gen} \in [0, 1]$ is the weight to choose between generating a word from the vocabulary or copying a word from the input sentence. We can find that the probability of word $w$ is the sum of probability of generating from the vocabulary and copying from the input sentence. If $w$ is an out-of-vocabulary word, $p^c(w)$ will be equal to 0; if $w$ is not in the input sentence, $\sum_{i=1}^{n} a^c_i$ will be equal to 0. In the training phase, the loss of decoder $L_{dec}$ at $c$-th step is calculated as follow:

$$L^{dec}_c = -log(P^c_{vocab}(\hat{w})), \tag{16}$$

where $\hat{w}$ is the target token at $c - th$ step. Then we calculate the overall loss as follows:

$$L^{dec} = -\frac{1}{G}\sum_{c=1}^{G} L^{dec}_c. \tag{17}$$

where $G$ is the length of target sequence. In the inference phase, the greedy search is utilized to decode and the target sequence is generated, which selects the token with the highest probability at step $c$ as the output token.

### Train and inference

During the training phase, the loss of contrastive learning task in the entity-type-aware module and the loss of entity recognition results generation task are jointly optimized. The total loss is calculated as follows:

$$L_{total} = \alpha L_{con} + (1 - \alpha)L_{dec}. \tag{18}$$

where $\alpha$ is the weight between the loss of contrastive learning task and the loss of entity recognition results generation task.

Inspired by *Athiwaratkun et al. (2020)*, during the training and inference phases, given an input text $X$, the target sequence is considered to convert entity spans in the $X$ to a span like "[entity span|entity type]". This approach of constructing the target sequence not only leverages the powerful text reconstruction and contextual understanding capabilities of the BART model, but also provides explicit contextual information during the text generation process, which can reduce the ambiguity in the process of generating text.

During the inference phase, the input text is fed into BART and the greedy search algorithm is utilized to directly generate the target sequence. After obtaining the target sequence, the entity recognition results are obtained by rule post-processing.

## EXPERIMENTS
### Setting
#### Datasets

Experiments are conducted on two legal NER datasets, CAIL2021 and Drug (*Chen et al., 2020*). CAIL2021 and Drug are split into train set, dev set, and test set in the ratio of 8:1:1. The statistics of these datasets are shown in Table 3.

The first dataset, CAIL2021, is derived from the "China AI and Law Challenge" and contains a wide variety of legal documents, including judgments from civil and criminal cases. This dataset is particularly rich in legal terminology and complex entity structures, making it ideal for testing the ability to recognize and classify detailed legal entities. It comprises over 2,000 annotated documents with entities labeled across several categories such as 'Defendant', 'Victim', 'Legal Article', and 'Crime Type'. The second dataset, known as 'Drug', focuses specifically on criminal cases related to drug offenses. It includes annotations for entities that are crucial for understanding the judicial processes in drug-related cases, such as 'Defendant', 'Drug Type', 'Sentence', and 'Arrest Location'. This dataset provides a narrower scope compared to CAIL2021, allowing us to test the adaptability of our model to specialized domains within the legal field. Both datasets are split into training, validation, and test sets with a distribution of 80%, 10%, and 10%, respectively, ensuring sufficient data for training and robust evaluation. The entities in these datasets are annotated with precise boundaries and types, providing a challenging yet realistic setting for assessing the performance of our NER framework. To introduce the

**Table 3 The statistics of CAIL2021 and Drug.**

| Dataset | Statistic | Train | Test | Dev |
|---|---|---|---|---|
| CAIL2021 | Items | 4,126 | 525 | 522 |
| | Chars | 262,956 | 34,158 | 32,446 |
| | Entities | 20,869 | 2,726 | 2,573 |
| Drug | Items | 1,200 | 176 | 177 |
| | Chars | 201,823 | 29,189 | 30,765 |
| | Entities | 12,853 | 1,877 | 1,910 |

**Table 4 The statistics of entity types in CAIL2021.**

| Entity type | Count | Proportion |
|---|---|---|
| Victim | 3,108 | 11.66% |
| Defendant | 6,463 | 24.24% |
| Stolen items | 5,781 | 21.68% |
| Organization | 806 | 3.02% |
| Value of item | 2,090 | 7.84% |
| Crime time | 2,765 | 10.37% |
| Crime location | 3,517 | 13.19% |
| Stolen currency | 915 | 3.43% |
| Crime tool | 735 | 2.76% |
| Profit | 481 | 1.80% |

statistics details of entity types in CAIL2021 and Drug we show the corresponding statistical results in Tables 4 and 5.

### Implementation details

Entity prediction is considered correct when both entity span and entity type are correctly predicted. Precision, recall, and F1 score are considered as our evaluation metrics. In our method, the BART-Large model (*Zhao et al., 2019b*) is used for experiments, with 12 layers in both the encoder and decoder. For a fair comparison, the pre-trained model with the same encoder and decoder layers as the BART-Large model is used in the comparative experiments. During the training phase, the batch size is set to 16, and the learning rate is set to 1e-5. Furthermore, the result generation in the training and inference phase is performed using the greedy search algorithm.

### Baselines

To explore the effect of our framework in NER task in the legal domain, we use the following models as baselines.

- **BiAffineNER** (*Yu, Bohnet & Poesio, 2020*) utilizes the biaffine mechanism to enhance the ability of model for locating entity spans.

**Table 5** The statistics of entity types in Drug.

| Entity type | Count | Proportion |
| --- | --- | --- |
| Defendant | 8,067 | 41.86% |
| Drug | 4,221 | 21.90% |
| Crime time | 2,693 | 13.97% |
| Crime location | 2,237 | 11.61% |
| Drug weight | 2,054 | 10.66% |

- **BERT-MRC** (*Li et al., 2020*) formulates the NER task as a machine reading comprehension task, and extracts possible text spans by answering the queries given the contexts.
- **BOCNER** (*Li et al., 2021*) combines the ordered neurons LSTM network (*Shen et al., 2018*) with the BERT model (*Devlin et al., 2019*) to tackle the NER task in the legal domain.
- **W$^2$NER** (*Li et al., 2022b*) models the NER task as word-word relation classification, and extracts entities by decoding based on the relation between word and word.
- **RoBERTa-BiLSTM-CRF** (*Shi et al., 2022*) combines the BiLSTM network with the RoBERTa model to tackle the Chinese NER task in the legal domain.
- **T5Based** (*Lee, Pham & Uzuner, 2022*) utilizes the T5 model to tackle the NER task in the financial text in a generative way.
- **PUnifiedNER** (*Lu et al., 2023*) is able to jointly train across multiple corpora, and implements in telligent on-demand entity recognition.
- **DiffusionNER** (*Shen et al., 2023*) formulates the named entity recognition task as a boundary-denoising diffusion process and thus generates named entities from noisy spans.

## Results and analysis

The overall performance of our method compared with the various baselines are shown in Tables 6 and 7. The experiments show that our framework outperforms the previous methods on CAIL2021 and Drug, which demonstrates the effectiveness of our proposed method in the NER task in the legal domain.

Concretely, compared with RoBERTa-BiLSTM-CRF, BOCNER, BiAffineNER, W$^2$NER and PUnifiedNER, our model can achieve better performance on CAIL2021 and Drug. RoBERTa-BiLSTM-CRF, BiaffineNER and BOCNER assist model in capturing the semantic feature of text, which can improve the performance of model in the NER task; owing to the ingenious design of the model, T5Based and W$^2$NER can solve the problem of intricate and lengthy entities in the legal texts, which lead to excellent performance in the NER task in the legal domain; PUnifiedNER is able to prove that the multi-dataset training empowered by prompting can also lead to significant performance gains over single-dataset training. However, they do not take better account of the semantic information of

**Table 6 The results of comparison among different models on CAIL2021.** Bold text indicates the best experimental results obtained in our study.

| Model | CAIL2021 | | |
| --- | --- | --- | --- |
| | Pr. | Rec. | F1 |
| BiAffineNER | 0.9124 | 0.8772 | 0.8944 |
| BERT-MRC | 0.9276 | 0.9182 | 0.9228 |
| BOCNER | 0.9021 | 0.9044 | 0.9032 |
| W$^2$NER | 0.9347 | 0.9206 | 0.9276 |
| RoBERTa-BiLSTM-CRF | 0.8889 | 0.9404 | 0.9139 |
| T5Based | 0.9255 | 0.9163 | 0.9208 |
| PUnifiedNER | 0.9286 | 0.9171 | 0.9228 |
| DiffusionNER | 0.9387 | **0.9253** | 0.9320 |
| **Ours** | **0.9414** | 0.9248 | **0.9330** |

**Table 7 The results of comparison among different models on Drug.** Bold text indicates the best experimental results obtained in our study.

| Model | Drug | | |
| --- | --- | --- | --- |
| | Pr. | Rec. | F1 |
| BiAffineNER | 0.8916 | 0.8587 | 0.8748 |
| BERT-MRC | 0.8935 | 0.8819 | 0.8876 |
| BOCNER | 0.8821 | 0.8736 | 0.8778 |
| W$^2$NER | 0.9083 | 0.8779 | 0.8929 |
| RoBERTa-BiLSTM-CRF | 0.8869 | 0.8723 | 0.8795 |
| T5Based | 0.8895 | 0.8816 | 0.8855 |
| PUnifiedNER | 0.8986 | 0.8753 | 0.8868 |
| DiffusionNER | 0.9107 | 0.8831 | 0.8967 |
| **Ours** | **0.9137** | **0.8859** | **0.8996** |

entity types and difference between each entity type, which leads to entity type prediction errors due to the diversity and professionalism of entity types in the legal text. By appropriately integrating entity type information into our model, we enhance its ability to learn semantic details of entity types and the representational differences among them, thereby improving the performance of our model. This approach demonstrates the critical importance of entity type information for NER task in the legal domain.

Compared with BERT-MRC, our model increases by 1.02% and 1.20% in terms of F1 score on CAIL2021 and Drug, respectively. Our model also performs slightly better than DiffusionNER. Similar to our method, BERT-MRC and DiffusionNER encode the comprehensive descriptions of entity type to the query to assist model in learning the information of entity types and improve the effect of model. While BERT-MRC and DiffusionNER incorporate the information of entity types during model training, their decoding process necessitates enumerating all possible entity segments. To maintain

efficiency, the length of these segments is often restricted and leads to suboptimal performance in recognizing intricate and lengthy entities within legal texts, thereby compromising the overall effectiveness of the entity recognition task. In contrast, our method incorporates an entity-type-aware module within the encoder, which not only aids in learning the information of entity types and the distinctions among different entity types but also utilizes a decoder equipped with a copy mechanism to directly generate results from entity extraction. This approach enables superior recognition of intricate and lengthy entities in legal texts. These factors help our method outperform BERT-MRC and DiffusionNER in legal NER task.

## Cross-validation discussion

In this experiment, we employed a $k$-fold cross-validation technique to enhance the accuracy of model performance assessments. Both the CAIL2021 and Drug datasets were divided into $k = 5$ subsets of similar size. For each iteration, one subset was designated as the validation set, while the remaining subsets served as the training set. This process was repeated across five rounds of training and validation. Furthermore, the model was executed five times for each fold (select runs are shown) to secure more robust statistical outcomes. Cross-validation results on the CAIL2021 dataset, the average performance metrics, obtained *via* cross-validation, are detailed in Table 8.

- Average Precision: $0.935 \pm 0.005$
- Average Recall: $0.920 \pm 0.004$
- Average F1 Score: $0.927 \pm 0.003$

The detailed performance metrics for selected runs and folds are presented in cross-validation results on the Drug dataset. The average performance metrics for the Drug dataset are summarized as follows:

- Average Precision: $0.910 \pm 0.006$
- Average Recall: $0.880 \pm 0.005$
- Average F1 Score: $0.895 \pm 0.004$

The specific metrics for selected runs and folds are illustrated in Table 9. Through comprehensive cross-validation and multiple executions per fold, the model consistently demonstrated stable performance across various data subsets, evidenced by minimal fluctuations in the performance metrics. The standard deviations of the precision, recall, and F1 scores underscore the reliability of our results, indicating robust generalization capabilities. This stability is essential for effectively handling named entity recognition tasks in the legal domain. Further comparisons with baseline models discussed within this article can substantiate the superior performance of our model.

## Ablation study

To further demonstrate the effectiveness of our method, we conduct the ablation study in the following three aspects:

**Table 8 Selected detailed performance metrics per fold and run on the CAIL2021 dataset.**

| Fold | Run | Precision | Recall | F1 Score |
|------|-----|-----------|--------|----------|
| 1 | 1 | 0.940 | 0.915 | 0.927 |
| 1 | 3 | 0.939 | 0.919 | 0.928 |
| 1 | 5 | 0.941 | 0.913 | 0.927 |
| 2 | 1 | 0.938 | 0.916 | 0.927 |
| 2 | 3 | 0.937 | 0.917 | 0.926 |
| 3 | 1 | 0.936 | 0.922 | 0.929 |
| 3 | 5 | 0.935 | 0.923 | 0.929 |

**Table 9 Selected detailed performance metrics per fold and run on the Drug dataset.**

| Fold | Run | Precision | Recall | F1 Score |
|------|-----|-----------|--------|----------|
| 1 | 1 | 0.915 | 0.875 | 0.895 |
| 1 | 3 | 0.912 | 0.876 | 0.894 |
| 1 | 5 | 0.916 | 0.877 | 0.896 |
| 2 | 1 | 0.913 | 0.878 | 0.895 |
| 2 | 3 | 0.914 | 0.879 | 0.895 |
| 3 | 1 | 0.915 | 0.874 | 0.894 |
| 3 | 5 | 0.916 | 0.877 | 0.895 |

## The effectiveness of each component

To validate the effectiveness of the entity-type-aware module and the copy mechanism in our method, they are individually removed from the model, and the final performance of the model is compared.

Tables 10 and 11 show the experimental results on CAIL2021 and Drug. The experimental results indicate that removing any component will have a negative impact on the performance of model. (i) If we remove the entity-type-aware module, the performance shows the biggest drop. The entity-type-aware module can help the model more accurately understand and process complex structures in legal texts through in-depth analysis and learning of the semantic features of different legal entity types. When the model is able to identify and understand the semantic differences between different entity types, it is more likely to accurately annotate key information in legal texts, thereby improving overall recognition accuracy. This proves that the entity-type-aware module has a positive impact on the performance of model. Meanwhile, the importance of entity type semantic information for the NER task in the legal domain is proved. (ii) Removing the copy mechanism also leads to a decline in model performance and indicates that the copy mechanism plays a crucial role in the text generation process. By constraining the text generated by the decoder, it ensures that the entities recognized by the model remain consistent with the content of the original legal texts, thereby positively impacting the overall effectiveness of the model.

**Table 10  The results of different components on CAIL2021.** Bold text indicates the best experimental results obtained in our study.

| | | CAIL2021 | | |
| --- | --- | --- | --- | --- |
| ETA | Copy mechanism | Pr. | Rec. | F1 |
| ✓ | × | 0.9391 | 0.9227 | 0.9307 |
| × | ✓ | 0.9380 | 0.9199 | 0.9288 |
| ✓ | ✓ | **0.9414** | **0.9248** | **0.9330** |

**Table 11  The results of different components on Drug.** Bold text indicates the best experimental results obtained in our study.

| | | Drug | | |
| --- | --- | --- | --- | --- |
| ETA | Copy mechanism | Pr. | Rec. | F1 |
| ✓ | × | 0.9113 | 0.8826 | 0.8967 |
| × | ✓ | 0.9096 | 0.8779 | 0.8934 |
| ✓ | ✓ | **0.9137** | **0.8859** | **0.8996** |

To determine the efficacy of the ETA module in mitigating entity type prediction errors, the entity recognition results for each entity type in CAIL2021 and Drug are analyzed. Tables 12 and 13 show the experimental results on CAIL2021 and Drug. The experimental results show that removing the ETA module leads to a decrease in term of F1 value for each entity type, which demonstrates the efficacy of the ETA module in addressing entity type prediction error.

### The impact of $\alpha$ in the multi-task learning

To explore the impact of different weights $\alpha$ in multi-task learning on the model's effectiveness, the model is trained with different weights $\alpha$, and the results are shown in Fig. 3.

As is shown in Fig. 3, we find that: (i) When $\alpha$ is equal to 0.05, 0.1 and 0.15 on CAIL2021, and when $\alpha$ is greater than 0.1 on Drug, the effect of model is better than that in the case of $\alpha$ equal to 0. (ii) When $\alpha$ is greater than or equal to 0.2 on CAIL2021, the effect of model is worse than that in the case of $\alpha$ equal to 0. We speculate that when $\alpha$ is set to an appropriate value, the contrastive learning task in the entity-type-aware module can not only assist the model in learning semantic information of entity types and the difference between each entity type, but also will not negatively influence the understanding and generation ability of the model, which makes the effect of the model have an overall improvement. On CAIL2021, when $\alpha$ is too large, the model assigns too much weight to the contrastive learning task during the training process, which negatively affects the optimization of the entity recognition results generation task, and influences the model's ability to understand the semantic information and generate the text. Consequently, it leads to the performance degradation of the model. The experimental results on Drug show that when the $\alpha$ value is low, the model may be too biased towards the entity

**Table 12 The recognition result for each entity type in CAIL2021.** Bold text indicates the best performing outcomes from our experiments.

| Entity type | Ours w/o ETA F1 | Ours F1 |
|---|---|---|
| Victim | 0.9370 | **0.9411** |
| Defendant | 0.9682 | **0.9707** |
| Stolen items | 0.9019 | **0.9112** |
| Organization | **0.9450** | 0.9347 |
| Value of item | 0.9778 | **0.9803** |
| Crime time | 0.9712 | **0.9734** |
| Crime location | 0.8566 | **0.8595** |
| Stolen currency | 0.9093 | **0.9168** |
| Crime tool | 0.8672 | **0.9059** |
| Profit | 0.9090 | 0.9090 |

**Table 13 The recognition result for each entity type in Drug.** Bold text indicates the best experimental results achieved in our study.

| Entity type | Ours w/o ETA F1 | Ours F1 |
|---|---|---|
| Defendant | 0.9520 | **0.9542** |
| Drug | 0.9367 | **0.9389** |
| Crime time | 0.9063 | **0.9139** |
| Crime location | 0.8620 | **0.8706** |
| Drug weight | 0.8704 | **0.8777** |

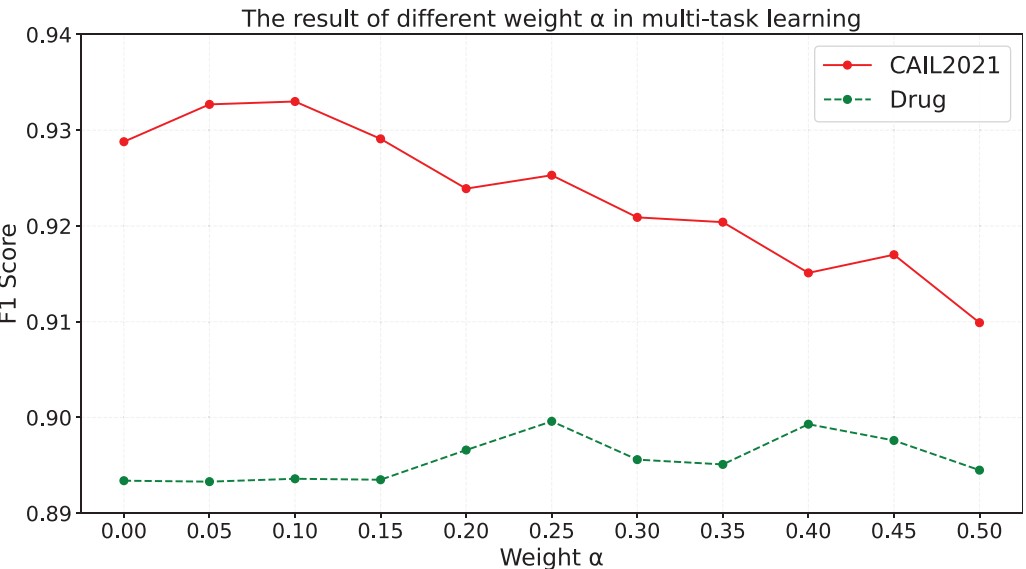

**Figure 3 The results of different weight $\alpha$ in multi-task learning.**

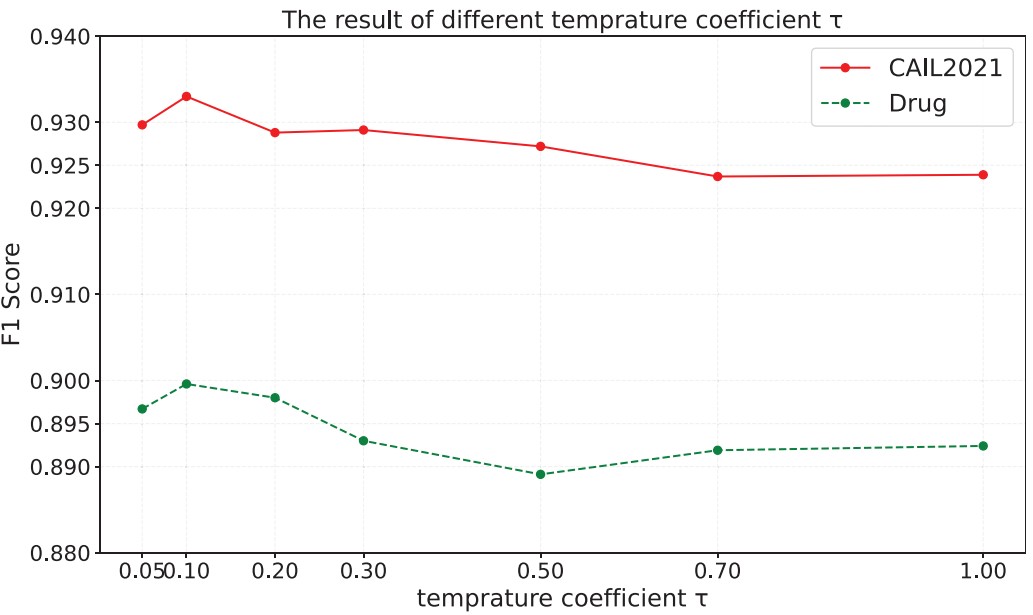

**Figure 4 The results of different temperature coefficient.**

recognition results generation task, and ignores the information gain that may be brought by the contrastive learning task, resulting in the F1 value not being significantly improved; while when the $\alpha$ value is gradually increased from 0.15 to 0.25, the model is able to reasonably balance the entity recognition results generation task and the contrastive learning task, and make full use of the two complementary advantages to achieve performance improvement. However, too high $\alpha$ values (*e.g.*, when $\alpha$ exceeds 0.25) may cause the model to consider the contrastive learning task too much and sacrifice the performance of the entity recognition results generation task, resulting in a fluctuating trend of high and low F1 values. As is shown in Fig. 3, the model achieves the optimal performance on the CAIL2021 when $\alpha$ is set to 0.1, whereas setting $\alpha$ to 0.25 yields the best performance on the Drug dataset.

### Temperature coefficient $\tau$ in the contrastive learning task

Temperature coefficient $\tau$ is a important parameter in the contrastive learning task, because it can affect the shape of the logits distribution and model's discrimination against negative samples. To explore the influence of different temperature coefficient $\tau$ in the contrastive learning task on the effect of the model, we train the model with different temperature coefficient $\tau$, and the results on CAIL2021 and Drug are shown in Fig. 4.

It is obvious that when $\tau$ is set to a smaller value, the model's performance is significantly better than that when $\tau$ is set to a larger value. A possible reason is that a smaller temperature coefficient makes the model understand and utilize hard negative samples more effectively, so it is easier to form a uniform representation space, which leads to better performance of the model. Therefore, the performance of the model is greatly influenced by the setting of the temperature coefficient, highlighting the importance of setting an appropriate coefficient for each dataset.

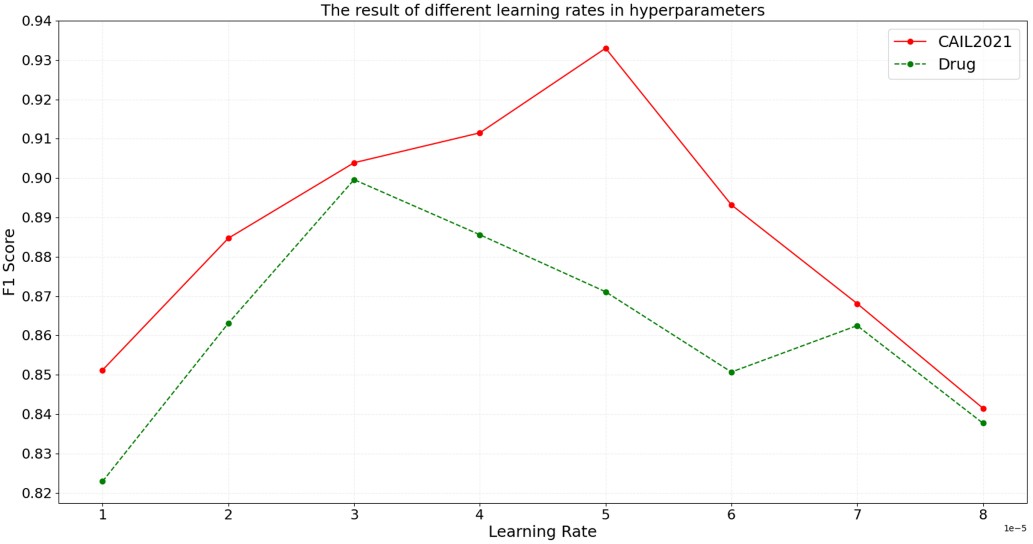

**Figure 5  The result of different learning rates in hyperparameters.**

### Learning rate in hyperparameters of the BART model

In the BART model, the learning rate parameter is critical to model training. The learning rate determines how much the model weights are adjusted, *i.e.*, how quickly the model responds to the loss function in each iteration. To assess the impact of the learning rate on model performance, the model is trained with different learning rates, and the results on CAIL2021 and Drug are shown in Fig. 5.

As shown in Fig. 5, on the CAIL2021 dataset, the optimal learning rate that achieves the best model performance is 5e-5, at which the F1 score peaks. When the learning rate is below this value, the model's F1 score progressively increases, suggesting that a lower learning rate enhances the model's ability to capture intricate details in legal texts. Conversely, when the learning rate exceeds 5e-5, the F1 score gradually decreases, which suggests that a high learning rate might cause oscillations during model training, adversely affecting convergence and performance. On the Drug dataset, the optimal learning rate is 3e-5. This dataset exhibits higher sensitivity to learning rate changes; a slight increase to 3e-5 significantly reduces performance, demonstrating a strict need for precise learning rate selection. The experimental results show that the moderate learning rate can better balance the model training speed and performance, and avoid the problem of overfitting and underfitting.

### Batch size in hyperparameters of the BART model

Batch size, referring to the number of samples processed per training iteration, directly influences the gradient estimation and memory usage of the BART model. An optimal batch size ensures more stable gradient estimations, and facilitates faster model convergence while efficiently leveraging the parallel processing capabilities of modern hardware. To assess the impact of batch size on model performance, the experiments are conducted across different batch sizes, with the results illustrated in Fig. 6.

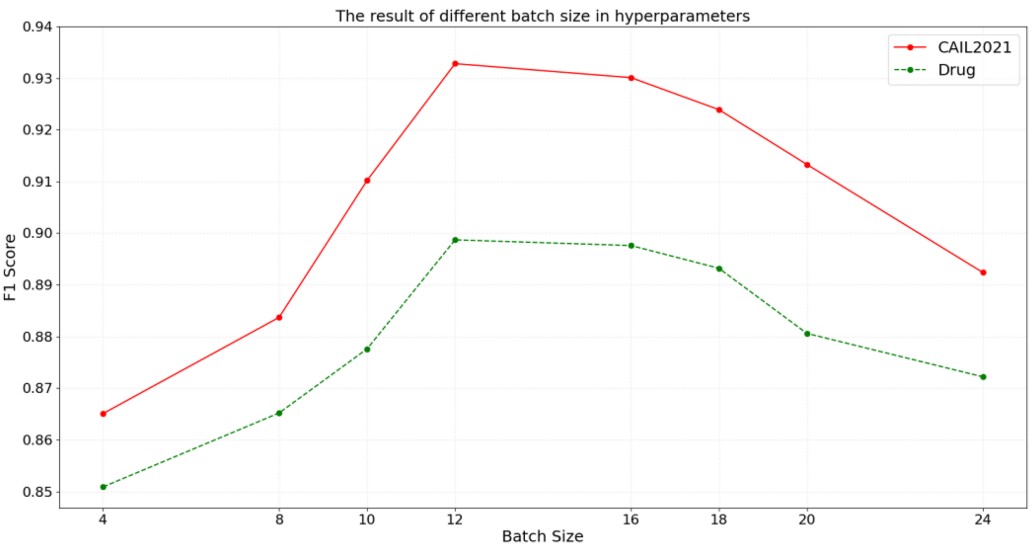

**Figure 6 The result of different batch size in hyperparameters.**

According to the results in Fig. 6, on the CAIL2021 dataset, the F1 score initially increases with the batch size but starts to decline after reaching a peak at a batch size of 12, demonstrating optimal data feature capture and generalization at this size. However, as the batch size increases to 24, the F1 score gradually decreases, possibly due to reduced exposure to diverse data per training cycle, which could hinder the model's ability to learn from a broader data distribution. Similarly, the Drug dataset shows an increase in F1 score with larger batch size, peaking at 12 before beginning to decrease. The above results suggest that while the smaller batch size can increase model update frequency, it may lead to the larger fluctuation during training. Conversely, overly large batch size might impair the model's generalization capability and training efficiency. Therefore, appropriately configuring batch size can accelerate model convergence while maintaining balance between generalization performance and training stability, thereby achieving the optimal outcomes in legal domain.

### Top-p in hyperparameters of the BART model

When using the BART model for named entity recognition, the top-$p$ parameter is a crucial factor that governs diversity in generation by determining which words are considered as potential outputs once the cumulative probability reaches a certain threshold. An array of top-$p$ settings ranging from 0.50 to 0.85 are tested on CAIL2021 and Drug. Detailed experimental results are shown in Fig. 7.

According to the results shown in Fig. 7, the F1 score on the CAIL2021 dataset initially increases with top-$p$ and reaches the optimal performance at a top-$p$ of 0.70. Beyond this point, as top-$p$ continues to increase, the F1 score begins to decrease, possibly due to the introduction of excessive irrelevant information, which reduces the model's precision. Similarly, the Drug dataset shows a peak F1 score at a top-$p$ of 0.65, with performance declining thereafter due to noise. The above results suggest that the selection of top-$p$

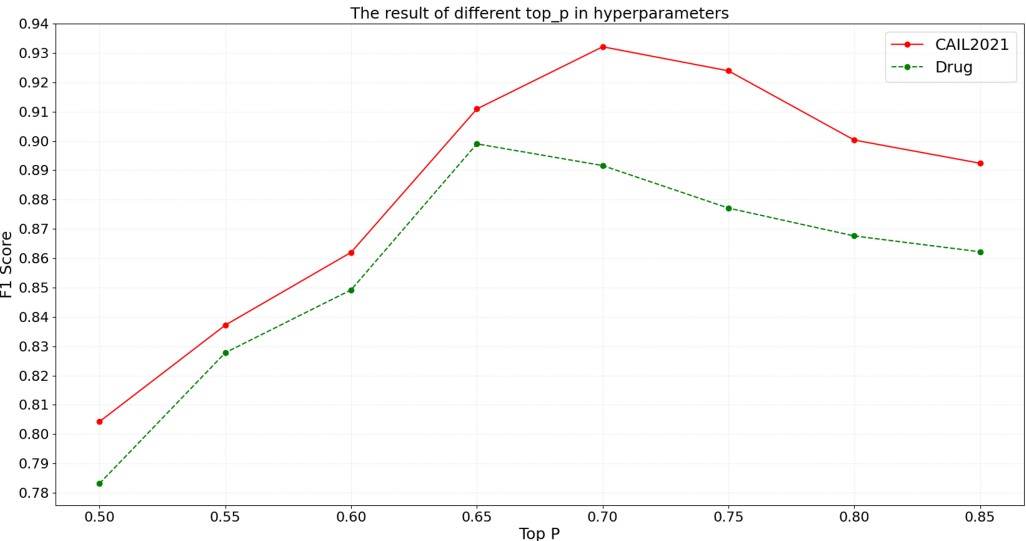

**Figure 7 The result of different top-$p$ in hyperparameters.**

should be optimized based on the specific characteristics of the dataset; such a low top-$p$ may limit the model's learning capacity, while such a high value can lead to information overload, affecting the model's accuracy and generalizability.

### Dropout rate in hyperparameters of the BART model

Dropout, as an effective regularization technique, helps to mitigate overfitting and enhance model generalization on unseen data. By conducting tests with eight different dropout rates on the CAIL2021 and Drug datasets, the specific impact on the F1 score is observed. The detailed experimental results are shown in Fig. 8.

According to the results shown in Fig. 8, the F1 score on the CAIL2021 dataset initially increases with the rise in dropout rate, peaking at 0.15, before declining with further increases. Similarly, on the Drug dataset, the highest F1 score occurs at a dropout rate of 0.10, with performance dropping thereafter. The above results suggest that the optimal dropout rate can help judicial named entity recognition models balance model complexity and noise handling, preventing overfitting. However, such a low dropout rate might not sufficiently deactivate neurons randomly during training, potentially leading to overfitting, while such a high dropout rate could lead to excessive information loss, impairing the model's ability to capture key features.

### Analysis of intricate and lengthy entity recognition

To illustrate the performance of the model on intricate and lengthy entities, the entities are divided into three groups according to their lengths. Our model is compared with W$^2$NER, which models the NER task as word-word relation classification and enables intricate and lengthy entity recognition. Tables 14 and 15 show the experimental results. Compare with W$^2$NER, our model increases by 0.31% and 5.35% in term of F1 for intricate and lengthy entities ($L > 10$) on CAIL2021 and Drug, respectively. A possible explanation is that

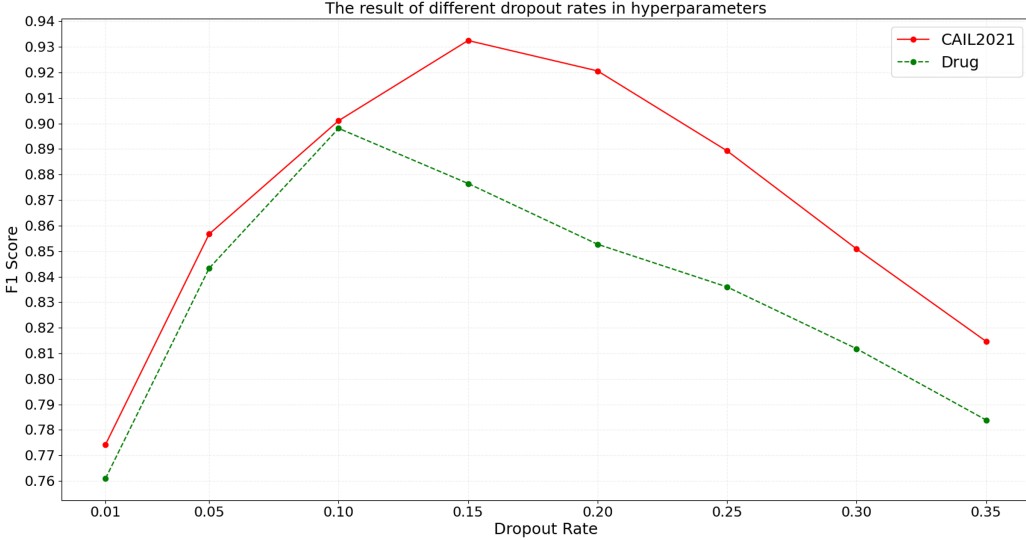

**Figure 8 The result of different dropout rates in hyperparameters.**

**Table 14 A comparison of recognition F1 score on entities of different lengths on CAIL2021.** We divide the entities into three groups: $1 \leq L \leq 5$, $5 < L \leq 10$, and $L > 10$, where $L$ is the length of the entity.

| | CAIL2021(F1) | | |
|---|---|---|---|
| **Model** | **$1 \leq L \leq 5$** | **$5 < L \leq 10$** | **$L > 10$** |
| W$^2$NER | 0.9265 | 0.9358 | 0.9310 |
| **Ours** | **0.9313** | 0.9358 | **0.9341** |

**Table 15 A comparison of recognition F1 score on entities of different lengths on Drug.** We divide the entities into three groups: $1 \leq L \leq 5$, $5 < L \leq 10$, and $L > 10$, where $L$ is the length of the entity.

| | Drug(F1) | | |
|---|---|---|---|
| **Model** | **$1 \leq L \leq 5$** | **$5 < L \leq 10$** | **$L > 10$** |
| W$^2$NER | 0.9381 | 0.8207 | 0.8080 |
| **Ours** | 0.9308 | 0.8138 | **0.8615** |

W$^2$NER extracts entities by decoding based on the relation between word and word, which may lead to error propagation problem and affect its effect of recognizing intricate and lengthy entities. The better performance of our model demonstrates that our model is more effective in recognizing intricate and lengthy entities.

## Analysis of the scalability of the BART model

In the ideal scenario, as data scale increases, the model's inference time should grow linearly or even less. If experimental results demonstrate that processing time increases more slowly than data scale, this indicates good scalability of the model. To more

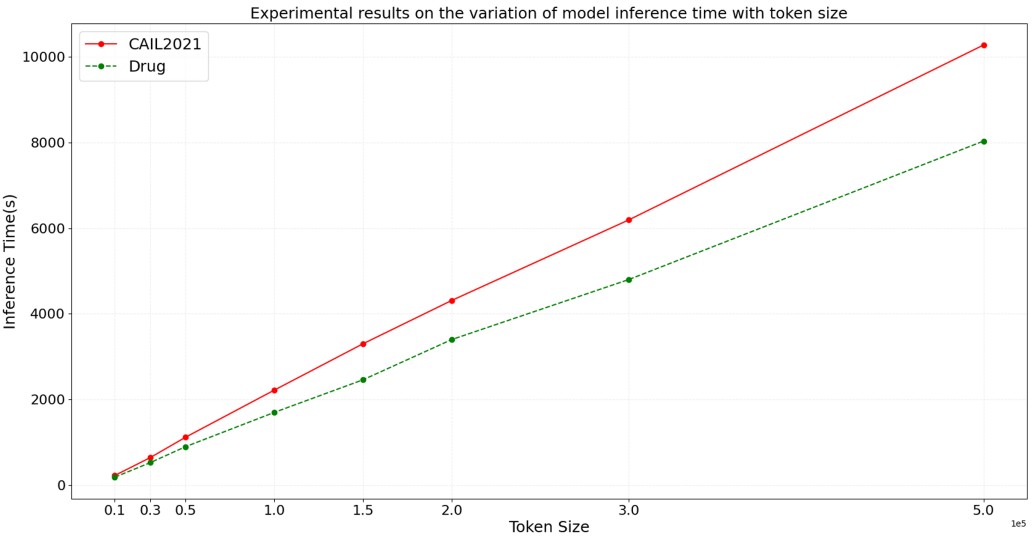

**Figure 9 Experimental results on the variation of model inference time with token size.**

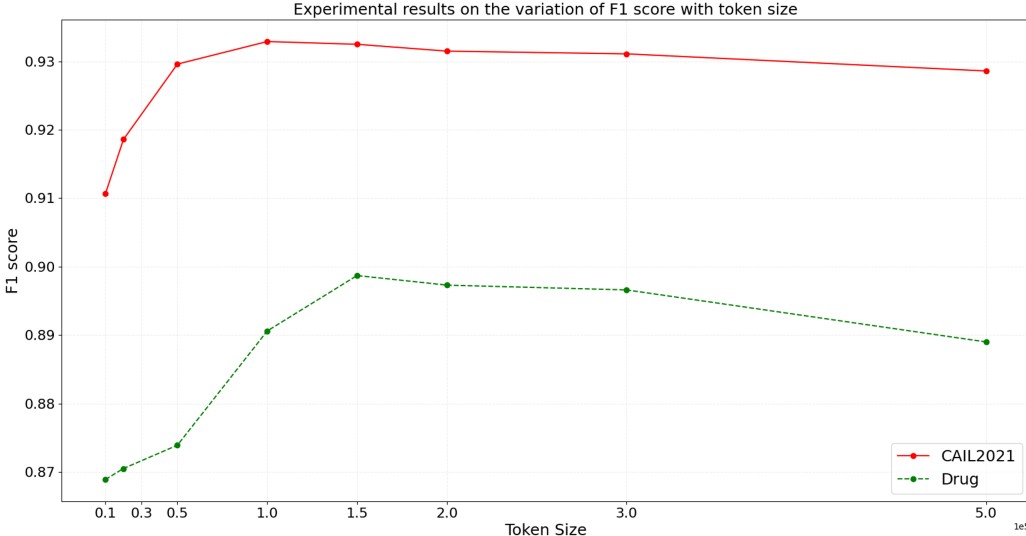

**Figure 10 Experimental results on the variation of F1 score with token size.**

accurately assess the scalability of the BART model, we design experiments with datasets of varying sizes from small to large scale. We then conduct inference using the BART model on these datasets, and the variation of inference time and F1 score with token size is shown in Figs. 9 and 10, respectively.

From the results depicted in Figs. 9 and 10, it is observable that on the CAIL2021 and Drug datasets, inference time increases with the number of tokens across eight different dataset scales, but not strictly linearly. The rate of increase in inference time is slightly slower than the rate of token increase, indicating that the BART model scales well with large datasets. For the performance variation, as data scales up, the model's F1 score

**Table 16 Case study.** We mark the correct entity boundaries and entity types with underlining and the entity boundaries and entity types predicted by traditional methods in italics.

| Case | Sentence | Our result |
|------|----------|-----------|
| Case 1 | The defendant Wang Mou stole an Iphone worth 5,000 yuan, and then sold the phone for [[*2,000 yuan*]_{value of item}]_{profit}. | The defendant Wang Mou stole an Iphone worth 5,000 yuan, and then sold the phone for [2,000 yuan \| profit]. |
| Case 2 | The defendant Zhang Moumou contacted He Moumou in advance, and sold about 3 g of methamphetamine drugs at a price of 700 yuan to He Moumou in [a game machine room on the west side of the intersection of [*Lanxi Road and Beishi Road, Putuo District, Shanghai*]_{location}]_{location}. | The defendant Zhang Moumou contacted He Moumou in advance, and sold about 3 g of methamphetamine drugs at a price of 700 yuan to He Moumou in [a game machine room on the west side of the intersection of Lanxi Road and Beishi Road, Putuo District, Shanghai \| location]. |

slightly improves initially and then begins to decline slightly. This trend might reflect the model's enhanced learning and generalization capabilities on medium-sized datasets, but as the dataset size increases, so do the challenges, possibly causing a slight performance decline. This performance dip could be also related to increased noise in larger datasets, changes in entity distribution, or more complex contextual relationships. Hence, the BART model demonstrates good scalability and effective handling of varying sizes of judicial named entity recognition datasets. Despite the increase in inference time with larger data scales, the sub-linear rate of time growth shows efficient large dataset handling, and signifies robust scalability. Performance variations across different data sizes show some fluctuations but remain relatively stable, suggesting that the BART model provides consistently stable performance, even on large-scale datasets.

### Case study

Some cases are illustrated in Table 16 to demonstrate that entity type prediction errors can be effectively reduced, and intricate and lengthy entities in legal texts can be accurately extracted by our model. In case 1, with the assistance of an entity-type-aware module to learn rich semantic information about entity types and the differences between each entity type, the entity type can be correctly assigned for the entity span by our model. In case 2, intricate and lengthy entities in the legal text can be accurately extracted by our model through the generation-based extraction method with the copy mechanism.

## CONCLUSIONS

In this article, we introduced a sequence-to-sequence framework specifically tailored for the named entity recognition (NER) task within the legal domain. By framing NER as a sequence generation problem and incorporating an entity-type-aware module, our approach significantly reduces entity type prediction errors. The framework's generative approach to entity extraction enhances the recognition of intricate and lengthy legal entities, ensuring detailed and accurate legal text processing. Experimental results validate the effectiveness of our method, demonstrating substantial improvements in both precision and recall. Our framework not only outperforms existing state-of-the-art models but also shows a marked increase in F1 score across two challenging legal datasets. This underscores our method's capability to handle the complexities and unique requirements

of legal documents, positioning it as a robust tool for legal NER tasks. Furthermore, our experiments confirm that the framework is adaptable and scalable, capable of maintaining high performance even as the complexity of the data increases. This makes it a valuable asset for ongoing and future implementations in legal informatics, where accuracy and detail are paramount.

### Funding
This research was supported by the National Natural Science Foundation of China [62172449, 72374070], Hunan Provincial Natural Science Foundation of China [2022JJ3021], Training Program for Excellent Young Innovators of Changsha [kq2107004], The science and technology innovation Program of Hunan Province [2022RC1105], and Industry-University-Research Innovation Fund of Chinese University [2021ITA01023]. This work was supported by the High Performance Computing Center of Central South University. The funders had no role in study design, data collection and analysis, decision to publish, or preparation of the manuscript.

### Grant Disclosures
The following grant information was disclosed by the authors:
National Natural Science Foundation of China: 62172449,72374070.
Hunan Provincial Natural Science Foundation of China: 2022JJ3021.
Training Program for Excellent Young Innovators of Changsha: kq2107004.
The science and technology innovation Program of Hunan Province: 2022RC1105.
Industry-University-Research Innovation Fund of Chinese University: 2021ITA01023.
High Performance Computing Center of Central South University.

### Competing Interests
The authors declare that they have no competing interests.

### Author Contributions
- Xingliang Mao conceived and designed the experiments, authored or reviewed drafts of the article, and approved the final draft.
- Jie Jiang conceived and designed the experiments, authored or reviewed drafts of the article, and approved the final draft.
- Yongzhe Zeng performed the experiments, performed the computation work, prepared figures and/or tables, and approved the final draft.
- Yinan Peng performed the experiments, performed the computation work, prepared figures and/or tables, and approved the final draft.
- Shichao Zhang analyzed the data, authored or reviewed drafts of the article, and approved the final draft.
- Fangfang Li analyzed the data, authored or reviewed drafts of the article, and approved the final draft.

## Data Availability

The raw data and experimental code are available in the Supplemental Files.

CAIL2021-Judicial Text Information Extraction from China Legal Intelligence Technology Evaluation is available at GitHub: https://github.com/china-ai-law-challenge/CAIL2021/tree/main/xxcq.

The CAIL2021 dataset is open sourced by the organizer of CAIL in https://github.com/china-ai-law-challenge/CAIL2021/tree/main/xxcq.

The Drug dataset is open sourced by the Information Retrieval Laboratory of Dalian University of Technology in https://github.com/china-ai-law-challenge/CAIL2021/tree/main/xxcq.

## Supplemental Information

Supplemental information for this article can be found online at http://dx.doi.org/10.7717/peerj-cs.2428#supplemental-information.

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
