# Peer review of "Generative named entity recognition framework for Chinese legal domain"

_PeerJ Computer Science, doi:10.7717/peerj-cs.2428_

## Round 0.1 · original submission · Minor Revisions

Dear authors,

Thank you for your submission. Feedback from the reviewers is now available. Your article has not been recommended for publication in its current form. However, we do encourage you to address the concerns and criticisms of the reviewers and resubmit your article once you have updated it accordingly. Furthermore, equations in the manuscript should be mentioned and used with correct equation number.

Best wishes,

Reviewer 1 ·

Basic reporting

Introduction part need to mention clearly need of NER for legal documents under consideration.
Potential application areas of problem statement under consideration need to be added

Experimental design

in the given article there is a mention of few entity types . It is not clear whether authors are considering only those entity classes from existing datasets or some additional one

Validity of the findings

Results and findings are clearly stated and supported with appropriate visuals.

Additional comments

The article explains all the techniques used with clear understanding and results are discussed well with respect to two established datsets

Cite this review as

Reviewer 2 ·

Basic reporting

- The English is quite clear,
- The literature selected is relevant and well referenced,
- Table 1 and table 2 have the same definition for some entities so the explanation becomes redundant.

Experimental design

- The content of the article is still in accordance with the objectives and scope of the journal,
- The method has been explained in detail,
- It would be more complete if the data Preprocessing could be explained briefly even though the data used takes or refers to 2 existing data,
- The evaluation process is explained briefly and would be more complete if it could be explained more completely,
- The model selection method has been explained adequately,

Validity of the findings

- The experiments and evaluations have been carried out well,
- The conclusions can be more complete if supplemented with information on the results of the experiment, for example the average difference in the comparison of its advantages.

Additional comments

In general, this paper is written in good language, has an interesting background and objectives. Nevertheless the motivation is still shallow. It mentioned in the abstract, the motivation was the predicting entity boundaries in NER leads to more errors since legal texts contain intricate and lengthy entities, but what kind of errors? It is good you give an example in the introduction section, I think the similar problem may also occur in the non-legal NER, so it will better if you compared to non-legal text NER, please give example legal vs non-legal NER. Following are several things that can be added to make this paper more complete, including:
- in the abstract should be explicitly stated the results
- please give more arguments why the sequence-to-sequence is appropriate for the legal NER since it also applied to non-legal NER
- Table 1 and table 2 have redundant definition information, so it is recommended to use one more complete table, and add an explanation or special mark in the table for the special entity used in this paper.
- The dataset used in this study was taken from a previous research dataset, so it is recommended to add an explanation in the dataset sub-chapter to make it more complete.
- The evaluation process has only been explained briefly, so it is recommended to provide a more complete explanation.
- The conclusion can be more complete if it is added with information on the results of the experiment, for example the average difference in the comparison of its advantages.

Cite this review as

Reviewer 3 ·

Basic reporting

The authors propose a generative framework for legal NER tasks in Chinese documents, using the seq2seq technique, using the pre-trained BART model as a baseline. The proposal considers the semantic particularities of legal entities, used in the accuracy of complex entities, usual in documents in the legal context.

The contributions of the research are clear (lines 67 to 74), and can be summarized as (1) use of a sequence-to-sequence model, (2) a module that improves entity type prediction through contrastive learning, and (3) use of a copy mechanism in the decoder, improving the accuracy in identifying complex and extensive entities.

Experimental design

Several experiments were conducted with the proposed framework on two pre-existing legal corpora, with results compared with eight state-of-the-art NER models. In addition, an ablation study was conducted on the model used to validate the effectiveness of the proposed modules.

The results demonstrated that the proposed model outperforms state-of-the-art methods in entity identification (Tables 7 to 10). In the ablation study, the results of the proposed techniques individually ("entity-type-aware" and "copy mechanism") demonstrated that the best result was obtained using the combination of the techniques (Tables 9 and 10), emphasizing the effectiveness of the proposal, both at a general level and in the individual analysis by entities (Tables 11 and 12), which demonstrates that they are vital for the effectiveness of the model in the NER task.

Validity of the findings

The results presented in the research demonstrated that the framework was effective in resolving errors in predicting entity types, improving the accuracy in identifying complex and long entities.

Additional comments

Regarding the text, the writing is well done, and the flow of ideas presented makes it easy to understand.
The research contributes significantly to the advancement of the state-of-the-art in legal NER by introducing a generative method that considers the semantic particularities of legal entities, which are notoriously difficult due to the complexity and specificity of the domain.

The integration of contrastive learning modules and copy mechanisms to directly address entities without the need for specific tagging schemes is not common in the state-of-the-art of legal NER, opening up space for future research. This approach not only improves prediction accuracy in legal contexts but can also be adapted to other specialized domains, offering a new direction for research in AI Applications

I believe that some suggestions could help to improve the analysis of the results:
- the research discusses conducting experiments to evaluate the performance of the proposed model, using several standard metrics for NER (precision, recall, and the F1 measure), as expected in the state-of-the-art.
- however, I did not find any information about the implementation of the cross-validation technique, the number of model runs, or the use of statistical measures (e.g., mean or standard deviation). Are the results presented in the tables from a unique run? Or the best result from N runs?
- for a more robust understanding of the model's performance and stability, I believe it would be beneficial to include information on whether or not the techniques mentioned above (cross-validation, mean, standard deviation, etc.) were used and, if so, to provide more details on these implementations.
- I believe that including this information could significantly increase the credibility and acceptance of the research results in the scientific community.
- I also believe that the scientific community would benefit if the supplementary material (codes, corpora, json files, etc.) were made publicly available on Github, for replication of results and future extensions of the studies.

Finally, I congratulate the authors for their extensive work and thank them for the scientific contribution they have made to the NER and NLP community.

Cite this review as

---

## Round 0.2 · accepted · Accept

Dear authors,

Two of the original reviewers did not respond to the invitation for reviewing your revised manuscript. The other referee thinks your paper can be accepted. I also think that the paper has been sufficiently improved. As such, the article is considered acceptable.

Best wishes,

Reviewer 2 ·

Basic reporting

no-comment

Experimental design

no-comment

Validity of the findings

no-comment

Additional comments

I appreciate your responses to my review, however, only one minor that I think you should add: please in the abstract give explicitly results quantitatively, not only in qualitative.

Cite this review as